# Towards Efficient SNNs: Sensitivity-Guided Pruning for Deep Spiking Architectures

## Abstract

Spiking Neural Networks (SNNs) offer compelling advantages in energy efficiency and biological plausibility but face performance and deployment challenges due to redundant structural units in suboptimal architectures. Existing compression techniques predominantly rely on unstructured connection-level pruning, which often necessitates specialized hardware for efficient execution. To overcome these limitations, we propose SPTE (Sensitivity-guided Pruning by Taylor Expansion), a structured pruning framework that leverages Taylor expansion to estimate each convolutional kernel's sensitivity to the loss function during training. This enables the iterative removal of less critical components. Extensive experiments across four benchmark datasets demonstrate the effectiveness of SPTE. Remarkably, SPTE achieves 78.09% connectivity sparsity on CIFAR10 with a +1.49% accuracy gain, outperforming previous state-of-the-art methods in both performance and model compactness.

## 1 Introduction

Spiking Neural Networks (SNNs) have emerged as a biologically inspired computational paradigm capable of processing spatiotemporal patterns with remarkable energy efficiency (Maass, 1997; Roy et al., 2019; Shaban et al., 2021). Unlike traditional Artificial Neural Networks (ANNs), SNNs employ discrete spike trains to enable event-driven communication, which offer significant potential for energy-efficient computation. This property also makes SNNs particularly attractive for deployment in neuromorphic hardware and in edge computing environments (Davies et al., 2018; Pei et al., 2019). Recent advancements in SNN training algorithms—such as surrogate gradient descent—have enabled deep SNNs with complex architectures to achieve competitive performance on various computer vision tasks (Wu et al., 2018; Fang et al., 2021; Ren et al., 2024). However, as SNN architectures grow deeper, they often suffer from excessive redundancy, particularly within spiking-based convolutional layers, resulting in suboptimal computational overhead and limited scalability.

To mitigate these challenges, pruning has emerged as a key strategy for compressing SNNs. Existing methods fall into two main categories: unstructured pruning and structured pruning. Unstructured pruning operates at the synaptic connection level, selectively removing individual weights deemed unimportant (Chen et al., 2021; Shen et al., 2023). While such methods effectively reduce parameter counts, they produce irregular sparsity patterns that require specialized hardware accelerators or customized software support to achieve practical runtime speedups, limiting their applicability in standard deployment pipelines. In contrast, structured pruning eliminates entire neurons or filters, yielding more hardware-friendly architectures and enabling straightforward acceleration on off-the-shelf computing platform. However, structured pruning remains relatively underexplored in the context of SNNs, primarily due to the inherent difficulty of quantifying structural importance under spiking dynamics (Ma et al., 2024). Recent studies (Li et al., 2024a;b) have proposed neuron- and channel-level pruning strategies based on spiking activities, yet these methods still lack a principled mechanism for explicitly measuring the contribution of individual spiking units. Consequently, the development of robust, contribution-aware structured pruning techniques for SNNs remain an open and promising research direction.

In Artificial Neural Networks (ANNs), sensitivity-based pruning strategies, especially those grounded in Taylor expansion, have demonstrated strong efficacy in identifying redundant parameters while preserving model performance (Molchanov et al., 2016; 2019). Approaches such as Tay-

lor Pruning utilize gradient- or Hessian-based approximations to estimate the saliency of individual units, providing a principled and theoretically grounded framework for structured pruning. These methods not only justify pruning decisions theoretically but also demonstrate strong empirical results across diverse architectures and tasks. However, such principled importance estimation mechanisms remain largely underexplored in the context of SNNs, as the discrete and non-differentiable nature of spiking activity introduces unique challenges in accurately measuring the contribution of spiking units. Motivated by these observations, we seek to extend contribution-aware analysis to the spiking domain, providing a foundation for informed and effective structural pruning in SNNs.

In this paper, we present Sensitivity analysis by Taylor Expansion (SPTE), a novel structured pruning framework for SNNs that leverages the precise contribution of spiking units. SPTE systematically eliminates entire convolutional kernels by estimating their contribution to the loss function via first-order Taylor expansion. Drawing inspiration from structured pruning methods in ANNs, SPTE provides a principled mechanism for reducing network complexity without compromising task performance. Unlike most ANN-based techniques, our method explicitly accounts for the unique characteristics of SNNs, primarily their temporal dynamics, ensuring compatibility and performance retention on multiple convolutional network architectures such as VGG and ResNet. The key contributions are summarized as follows.

- We propose SPTE, a Taylor-expansion-based sensitivity analysis framework for structured pruning in convolutional SNNs, which evaluates kernel-level importance of spiking units to guide informed pruning decisions.

- SPTE achieves superior trade-offs between accuracy and model compactness, with highlights such as +1.49% accuracy improvement and 78.09% connectivity sparsity on CIFAR10, and a +2.81% accuracy gain with 72.49% connectivity on Tiny-ImageNet, outperforming prior unstructured and attention-based methods.

- SPTE significantly reduces inference time and energy consumption while preserving classification accuracy, even for deep SNNs trained from scratch. It provides a principled, hardware-friendly, and scalable solution for optimizing SNN deployment in latency- and power-constrained environments.

## 2 RELATED WORK

### 2.1 SPIKING NEURAL NETWORKS

Spiking Neural Networks (SNNs) mimic the spatiotemporal dynamics of biological neurons and process information through asynchronous spikes (Maass, 1997). Early studies highlighted their potential for low-power, real-time applications such as sensory processing and robotics (Indiveri et al., 2011). However, training deep SNNs is challenging due to the non-differentiable nature of spike functions and temporal dependencies (Pfeiffer & Pfeil, 2018). Surrogate gradient methods and ANN-to-SNN conversion have helped scale SNNs to larger datasets (Neftci et al., 2019; Rueckauer et al., 2017), but these approaches often inherit the inefficiencies of their ANN counterparts, making model compression necessary (Roy et al., 2019).

### 2.2 MODEL COMPRESSION IN SNNS

Most SNN compression work has focused on unstructured pruning, quantization, and knowledge distillation (Shi et al., 2024; Wei et al., 2025; Zhang et al., 2023). Unstructured pruning methods, such as magnitude-based thresholding (Rueckauer & Liu, 2018) and Hebbian learning-inspired schemes (Diehl et al., 2015), remove individual weights based on their importance or activity. While effective in reducing parameter counts, these methods produce irregular sparsity patterns that are poorly supported by mainstream hardware accelerators (Han et al., 2016). Structured pruning, which eliminates entire neurons, channels, or filters, generates more regular and hardware-friendly models. However, structured pruning remains relatively underexplored in SNNs, largely due to the difficulty of quantifying structural importance in spiking systems where neuron activity is event-driven and temporally sparse (Zenke & Vogels, 2021).

## 2.3 SENSITIVITY ANALYSIS AND STRUCTURED PRUNING

In ANNs, sensitivity-based pruning methods, including those based on Taylor expansions (Molchanov et al., 2016; 2019), using Skeletonization to prune nodes (Mozer & Smolensky, 1988), learning to use agents to prune filters (Huang et al., 2018) and multi-armed bandits(MABs) (Ameen & Vadera, 2020), aim to model the impact of perturbation weights on the loss function. Extending these approaches to SNNs is challenging because spike generation is discrete and non-differentiable. Recent studies have adapted gradient-based pruning with surrogate gradients but are still limited to unstructured sparsity. In this work, we propose the SPTE method, which applies Taylor-based sensitivity metrics at the kernel level for convolutional SNNs. This structured pruning approach removes entire filters, producing compact, hardware-friendly models that maintain high accuracy with only minimal fine-tuning.

## 3 METHOD

This section introduces the proposed dynamic structural pruning framework of SPTE. This method automatically refines the structure of the neural network during training by evaluating the importance of channel through loss values. It works seamlessly with various convolutional network architectures like VGG, ResNet.

### 3.1 PRELIMINARY

**Spiking Neuron Model.** The spiking neuron model is the basic unit of spiking neural networks, which simulates the behavior of biological neurons transmitting information through spiking. The mechanism of action potential generation in biological neurons involves depolarization, overshoot, and repolarization processes. The spike neuron model simulates the mechanism of biological action potentials, thereby establishing a simplified mathematical model consisting of three equations: charging, discharging, and resetting. As shown in Eq.1, spike neurons accumulate input stimuli $X_t$ from presynaptic neurons and integrate all currents received from these neurons. If the membrane potential $H_t$ of a spiking neuron exceeds a certain threshold $V_{th}$, it will fire a spike and reset to the resting potential $V_{reset}$.

$$
\begin{aligned}
H_t &= f(V_t - 1, X_t) \\
S_t &= \Theta(H_t - V_{th}) \\
V_t &= H_t \cdot (1 - S_t) + V_{reset} \cdot S_t,
\end{aligned}
\tag{1}
$$

Where $\Theta(\cdot)$ represents the Heaviside step function, if a spike is fired, $S_t$ is 1, otherwise it is 0. The function $f(\cdot)$ is the equation that describes the dynamics of spiking neurons. In this paper, the spiking neuron models we use are the Integrate-and-Fire(IF) neuron model as presented in Eq.2 of Spikingjelly (Fang et al., 2023).

$$
V_t = f(V_{t-1}, X_t) = V_{t-1} + X_t
\tag{2}
$$

**Surrogate Gradient.** Unlike traditional neural networks, spiking neurons in SNN transmit feature information through a series of discrete spikes (Wu et al., 2018; Deng et al., 2022; Zhou et al., 2023). As shown in equation 3, backpropagation in SNN utilizes chain rules to calculate gradients across spatial and temporal dimensions, similar to the time backpropagation(BPTT) algorithm in recurrent neural networks (RNN).

$$
\frac{\partial L}{\partial W} = \sum_{t=1}^{T} \frac{\partial L}{\partial H_t} \frac{\partial H_t}{\partial W} = \sum_{t=1}^{T} \frac{\partial L}{\partial S_t} \frac{\partial S_t}{\partial H_t} \frac{\partial H_t}{\partial X_t} \frac{\partial X_t}{\partial W}
\tag{3}
$$

Due to the binary property of the spikes ($S_t \in \{0, 1\}$), $\frac{\partial S_t}{\partial H_t}$ is not differentiated. As shown in Eq.4, the gradient is positive infinity if x is 0, and 0 otherwise.

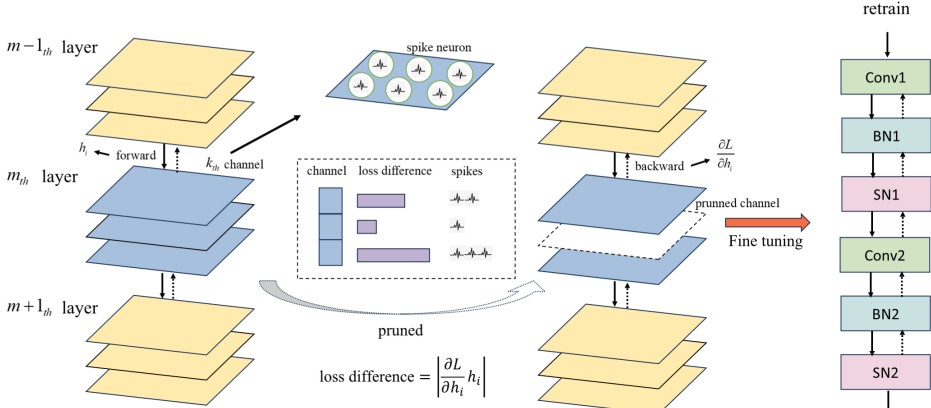

Figure 1: The schematic illustration of the SPTE structure framework. Calculate the loss difference for each channel and select the least one to prune it. Then, fine tune by retraining.

$$\delta(x) = \begin{cases} +\infty, & x = 0 \\ 0, & x \neq 0 \end{cases} \tag{4}$$

Therefore, to enable the use of backpropagation for the direct training of SNN, the gradient surrogate method employs differentiable functions $g(x)$ to substitute for not differentiated spikes $\Theta(x)$. The gradient surrogate function utilized here is the sigmoid function $g(x) = Sigmoid(\alpha x) = \frac{1}{1+e^{-\alpha x}}$, where $\alpha$ is used to control the smoothness of the function.

## 3.2 The Framework of SPTE

The inspiration comes from the plasticity of the biological nervous system, similar to how biological neural networks optimize information transmission through synaptic regulation, establishment, and elimination processes. Therefore, the SPTE structure framework simultaneously performs joint learning of network weight parameters and topology structure during the training process.The proposed method for pruning consists of the following steps: 1) Evaluate the importance of each channels to determine the sensitivity of pruning that channel to the network;2) Select the least important spiking neurons and remove them, namely topology structure learning;3) Fine tune it until it recovers or approaches its original performance, namely weight parameters learning.The framework is shown in Algorithm 1.

---

**Algorithm 1** The Overall Pruning Framework.

---

**Require:** Input dataset D, where X is data, Y is label.
**Ensure:** Pruning feature map ratio p%.
  1: **while** The ratio of pruned feature maps $< p\%$ **do**
  2:     **for** each convolutional layer of the network **do**
  3:         Calculate the activation value $h_i$ of each feature map during forward propagation;
  4:         Calculate the the gradient $\frac{\partial L}{\partial h_i}$ of each feature map during backward propagation;
  5:         Multiply $\frac{\partial L}{\partial h_i}$ and $h_i$, as the change in the loss function $\triangle L(D, h_i)$ ;
  6:     **end for**
  7:     Sort all feature maps by $\triangle L(D, h_i)$;
  8:     Prune the first q% of convolutional kernels;
  9: **end while**
10: **return** The lightweight SNN.

---

### 3.3 CHANNEL SENSITIVITY EVALUATION

In previous work, it has been proven that using first-order Taylor expansion holds for the following equation (Molchanov et al., 2016):

$$
\begin{aligned}
|\triangle L(D, h_i)| &= |L(D, h_i = 0) - L(D, h_i)| \\
L(D, h_i = 0) &= L(D, h_i) - \frac{\partial L}{\partial h_i} h_i + R_1(h_i = 0) \\
|\triangle L(D, h_i)| &= \left| L(D, h_i) - \frac{\partial L}{\partial h_i} h_i - L(D, h_i) \right| = \left| \frac{\partial L}{\partial h_i} h_i \right|,
\end{aligned}
\tag{5}
$$

where $D$ is a dataset, $h_i$ is the activation value of a feature map, $L(D, h_i = 0)$ is a loss value if the feature map $h_i$ is pruned and $L(D, h_i)$ is the loss if it is not pruned. $\triangle L(D, h_i)$ is the difference between two losses which we will use for evaluation. The second equation in Eq.5 is the first-order Taylor expansion of $L(D, h_i = 0)$.

Due to the shape of the input x is $T \times B \times C \times H \times W$, the shape of $L(D, h_i)$ is also $T \times B \times C \times H \times W$. In order to sort each feature map, we performed the following processing:

$$
I_c = \frac{1}{T \times B \times H \times W} \sum_{t=0}^{T-1} \sum_{b=0}^{B-1} \sum_{h=0}^{H-1} \sum_{w=0}^{W-1} |\triangle L(D, h_i)|_{t,b,c,h,w}
\tag{6}
$$

Now that we have evaluated the sensitivity of each feature map, we can proceed with sorting.

### 3.4 CHANNEL REMOVAL

To eliminate the redundant components in a neural network that contribute minimally to the target task or even introduce noise, we apply structured pruning by selectively removing a certain percentage of feature maps. This strategy not only reduces the model's computational complexity but also improves its generalization by discarding less informative or interfering channels. In SNN ResNet, deleting feature map from the previous convolutional layer may result in mismatched dimensions in the next layer. Therefore, we only prune the convolutional layers without skip connections.

Specifically, we determine the pruning targets based on the sensitivity of each feature map calculated before. The core idea is that channels with lower sensitivity have a smaller impact on the model's performance and can thus be safely removed.We start by setting a total pruning ratio p%, which represents the final sparsity level we aim to reach. To avoid severe accuracy drops caused by aggressive one-shot pruning, we prune iteratively: in each pruning iteration, we remove q% of the least important channels. In each iteration, the sensitivity scores of all channels are sorted in ascending order. The top q% are identified as the least important and thus selected for removal. Rather than setting the corresponding weights to zero, which would retain the original network structure and memory footprint, we physically remove the associated convolutional kernels. This process also involves shifting the remaining kernels forward in memory, ensuring structural compactness and efficiency. The calculation formulas for p is as follows:

$$
p\% = \frac{C_p}{C} \times 100\%
\tag{7}
$$

where $C_p$ is the number of pruned convolution kernels and $C$ is the total number of convolution kernels. The calculation of q is similar to that of p. In the experiment, we set q% to $\frac{1}{24}$.

Furthermore, this method enables real memory savings, unlike unstructured pruning, where sparse representations still require support from specialized libraries or hardware. By physically removing convolutional kernels, the model's architecture becomes inherently compact, simplifying downstream deployment without additional compression steps.

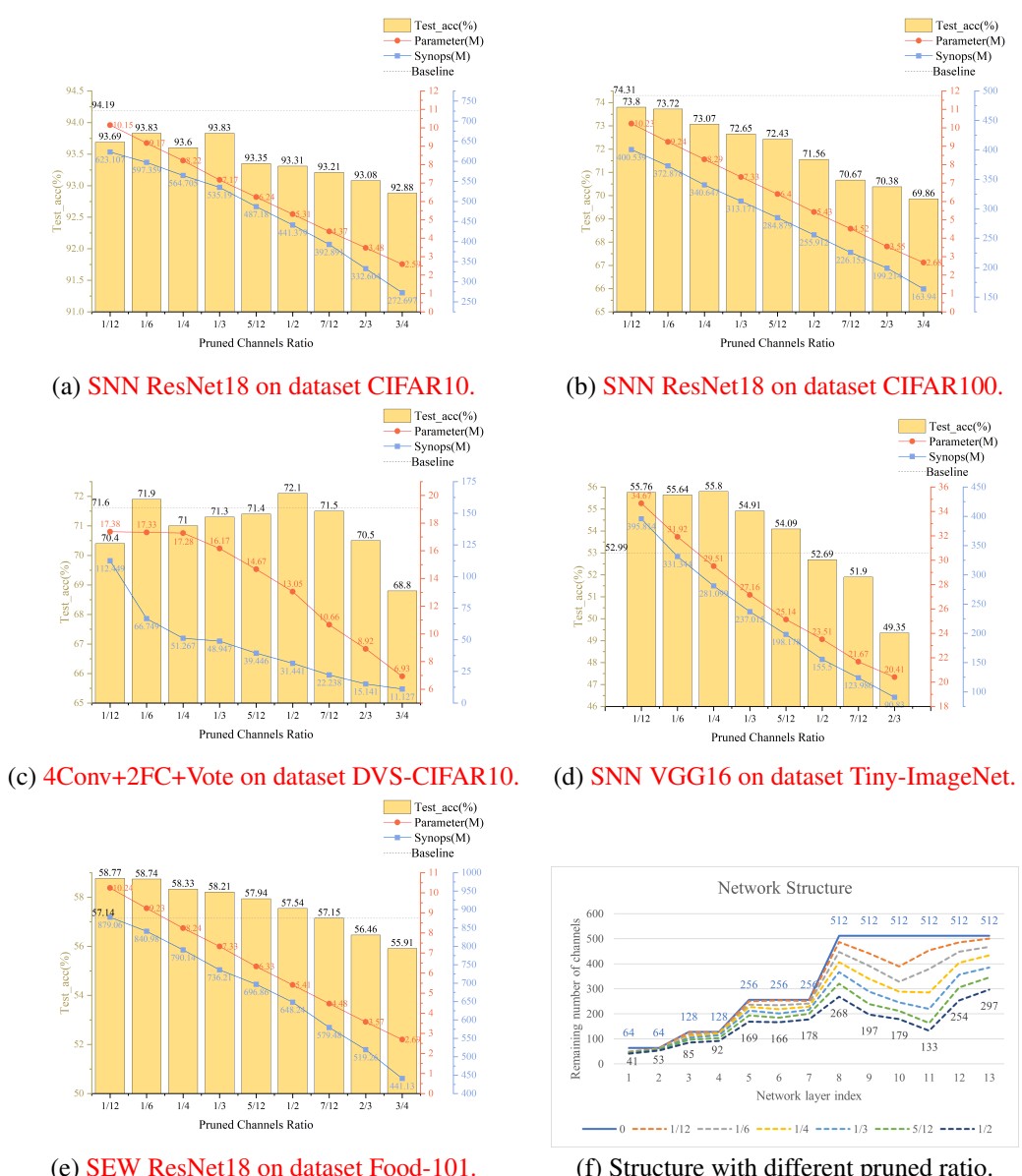

(a) SNN ResNet18 on dataset CIFAR10.

(b) SNN ResNet18 on dataset CIFAR100.

(c) 4Conv+2FC+Vote on dataset DVS-CIFAR10.

(d) SNN VGG16 on dataset Tiny-ImageNet.

(e) SEW ResNet18 on dataset Food-101.

(f) Structure with different pruned ratio.

Figure 2: The performance of the SPTE structure framework and network structure with different pruned ratio. The solid line is the structure of SNN VGG16 on dataset CIFAR10.

## 3.5 FINE TUNING

Suddenly disconnecting some neurons in a neural network, analogous to pruning synapses in the human brain, inevitably leads to a degradation in performance. This is because the structural integrity and learned representations of the network are disrupted when important parameters are removed. To mitigate this performance drop and allow the network to adapt to its new, sparser architecture, a fine-tuning phase is introduced immediately after each pruning step.

The fine-tuning process is essentially a retraining stage, where the model is optimized again using the same loss function and optimizer as in the original training, but typically with a reduced learning rate and fewer training epochs. The lower learning rate helps preserve the knowledge retained in the remaining parameters while gradually adapting the network to its pruned structure, thus avoiding overfitting or destabilization. This prune-and-fine-tune procedure is performed iteratively. In each iteration, the convolutional kernels in the network are pruned based on sensitivity. After each

Table 1: Inference speed before and after pruning.

| Dataset | Network | 0(it/s) | 1/2(it/s) |
|---|---|---|---|
| CIFAR10 | VGG16 | 13.86 | 15.55 |
| | ResNet18 | 10.15 | 12.44 |
| CIFAR100 | VGG16 | 26.50 | 33.73 |
| | ResNet18 | 20.40 | 22.93 |
| DVS-CIFAR10 | 4Conv+2FC +Vote | 13.45 | 19.45 |
| Tiny-ImageNet | VGG16 | 12.13 | 17.29 |

Table 2: Performance with different timesteps of SNN VGG16 on dataset CIFAR10.

| T | Ratio | Acc.(%) | Parameter(M) | FLOPs(G) |
|---|---|---|---|---|
| 2 | 0 | 91.27 | 33.64 | 193.63 |
| | 1/2 | 90.10 | 21.37 | 30.72 |
| 4 | 0 | 92.14 | 33.64 | 387.26 |
| | 1/2 | 91.14 | 21.37 | 60.88 |
| 8 | 0 | 90.58 | 33.64 | 774.51 |
| | 1/2 | 91.53 | 21.25 | 131.08 |

pruning phase, fine-tuning is applied to recover performance. Repeat these steps until the network reaches the target pruning rate p%. In our experiment, we set p% to $\frac{3}{4}$, which means a total of 18 iterations are performed (or stopped when the model performance is severely compromised), with 20 epochs of retraining per iteration. This iterative framework strikes a balance between model compactness and accuracy, gradually reducing redundancy while maintaining generalization ability.

## 4 RESULTS

The experiments were conducted using the SpikeJelly framework (Fang et al., 2020), an SNN library buit on PyTorch. Evaluation were performed on both static and neuromorphic datasets. For static evaluation, CIFAR-10 and CIFAR-100 are standard static image classification datasets containing 10 and 100 classes, respectively, with $32 \times 32$ RGB images. For SNN input, the images were encoded using 8 timesteps for CIFAR-10 and 4 timesteps for CIFAR-100. Tiny ImageNet contains 200 classes with 100,000 training samples and 10,000 testing samples of size $64 \times 64$, encoded with 4 timesteps. Food-101 contains 101 food categories with a total of 101000 images. Each category provides 250 manually reviewed test images and 750 training images. All images are scaled to a maximum edge length of 512 pixels. We resize to $224 \times 224$ encoded with 4 timesteps. For neuromorphic evaluation, the DVS-CIFAR10 dataset was used, with 9,000 training and 1,000 tesing samples. Event data were converted into frame representations using 4 timesteps. The network structures employed in the experiments were based on VGG and ResNet backbones. For the DVS-CIFAR10, the network structure is 128C3-MP2-128C3-MP2-128C3-MP2-128C3-MP2-FC2048-FC100-Vote10.

### 4.1 EVALUATION ON DIFFERENT DATASETS

As shown in Figure 2, the experimental results of the proposed framework are evaluated across multiple datasets. Different channel pruning ratios are applied while maintaining fixed step sizes for each training iteration. Model efficiency is evaluated using parameter count and SynOps, which serve as indicators of computational cost and energy consumption.

**Performance Analysis.** In Figure 2, the yellow bar charts represent test accuracy at different channel pruning ratios, while the line graphs show the number of parameters (red) and SynOps (blue), measured in M ($10^6$) simultaneously. It can be observed that as the channel pruning ratio increases, test precision experiences only a minor drop, while parameters and SynOps decrease significantly. In some cases, accuracy even improves at lower pruning ratios. These results highlight the effectiveness of our framework in compressing the network into a more lightweight model while preserving, and in some cases enhancing, predictive performance

On CIFAR-10, for the ResNet model in Figure 2(a), compressing the number of convolution kernels to about 7/12 results in less than 1% accuracy loss. On CIFAR-100, the ResNet model in Figure 2(b), compressing the number of convolution kernels by half reduces model size by more than 50% (original 11.22 M parameters) with only a 2.75% accuracy loss. For DVS-CIFAR10, as shown in Figure 2(c), reducing kernel by 1/6 and 1/2 increases accuracy by 0.3% and 0.5%, respectively. For Tiny ImageNet, as shown in Figure 2(d), accuracy slightly improves at most pruning ratios, and compressing to 1/2 results in about 0.3% accuracy drop. For Food-101, as shown in Figure 2(e),

Table 3: Comparison of experimental performance with other methods.

| Dataset | Method | Network | T | Top-1 Acc. (%) | Acc.Loss (%) | Avg.SOPs (M) | Param. (M) |
|---|---|---|---|---|---|---|---|
| CIFAR10 | Grad R (Chen et al., 2021) | 6Conv+2FC | 8 | 92.84 | -0.30 / -0.81 / -3.52 | 371.05 / 143.69 / 54.89 | 10.43 / 1.86 / 0.26 |
| | ESL-SNNs (Shen et al., 2023) | ResNet19 | 2 | 92.71 | -0.63 / -1.25 / -1.81 | 298.24 / 178.10 / 108.89 | 2.53 / 1.26 / 0.63 |
| | Unstruct-Pru (Shi et al., 2024) | 6Conv+2FC | 8 | 92.84 | -0.21 / -0.79 / -2.19 | 38.32 / 16.47 / 8.50 | 12.57 / 9.56 / 7.10 |
| | QP-SNN (Wei et al., 2025) | ResNet20 | 2 | 95.12 | -0.56 | - | 3.92 |
| | **SPTE** | ResNet18 | 8 | **94.19** | **-0.36 / -0.88 / -1.31** | **535.19 / 441.38 / 272.70** | **7.17 / 5.31 / 2.59** |
| CIFAR100 | Unstruct-Pru (Shi et al., 2024) | Resnet18 | 4 | 74.16 | -1.82 / -2.85 / -3.71 | 27.16 / 15.54 / 9.60 | 13.18 / 9.86 / 7.67 |
| | QP-SNN (Wei et al., 2025) | ResNet20 | 2 | 75.29 | -0.51 | - | 4.10 |
| | **SPTE** | Resnet18 | 4 | **74.31** | **-0.59 / -1.88 / -3.05** | **372.88 / 284.88 / 212.58** | **9.24 / 6.40 / 4.04** |
| DVS-CIFAR10 | ESL-SNNs (Shen et al., 2023) | VGGSNN | 10 | 82.4 | -1.8 / -3.2 / -4.9 | 266.17 / 152.64 / 79.65 | 1.94 / 0.97 / 0.48 |
| | Unstruct-Pru (Shi et al., 2024) | VGGSNN | 10 | 82.4 | -0.5 / -3.4 / -4.1 | 47.81 / 10.02 / 6.75 | 3.52 / 2.18 / 1.81 |
| | **SPTE** | 4Conv+2FC +Vote | 4 | **71.60** | **+0.3 / +0.5 / -0.1** | **66.75 / 31.44 / 22.24** | **17.33 / 13.05 / 10.66** |
| Tiny-ImageNet | SCA-based (Li et al., 2024b) | VGG16 | 4 | 49.33 | +0.03 / -0.19 | - / - | 27.92 / 19.76 |
| | QP-SNN (Wei et al., 2025) | VGG16 | 2 | 53.32 | -1.33 | - | 4.67 |
| | **SPTE** | VGG16 | 4 | **52.99** | **+2.77 / +1.1 / -1.09** | **395.81 / 198.18 / 123.99** | **34.67 / 25.14 / 21.67** |

When the pruning ratio is 1/12, the accuracy increases by 1.63%, then shows a slow downward trend. When the pruning ratio overtake 7/12, the accuracy is lower than that of the baseline model.

**Energy Consumption and Inference Speed Analysis.** During pruning, we quantify the reduction in energy consumption by calculating the number of Synaptic Operations(SynOps). As seen from the red line in Figure 2, with an increase in the network's channel pruning ratio, the synaptic operations gradually decrease. Meanwhile, to quantify inference speed, we calculated the number of iterations per second before and after pruning for each network to observe the comparison of model inference speed. The inference speed results are presented in Table 1, where 0 represents the original model and 1/2 denotes the trimmed model. It can be observed that all models exhibits improved inference speed after pruning. These indicates that our approach effectively reduces energy consumption and inference time, improves resource utilization, and enhances network performance, making it well-suited for deploying SNNs and achieving more efficient neural compution in resource-constrained environments.

## 4.2 SENSITIVITY DISTRIBUTION

To better analyze the importance of each layer within the network, we visualized the number of channels retained in each layer during the pruning process, as shown in Figure 2(f). Lines of different colors indicate the various pruning ratios, with the original model represented by a solid blue line and the pruned model shown as dashed lines in other colors. Using SNN VGG16 on the CIFAR-

Table 4: Comparison with ANN method.

| Method | PT? | Network | FLOPs reduction(%) | Acc.(%) | Acc.Loss (%) |
|---|---|---|---|---|---|
| CUP(Duggal et al., 2021) | × | ResNet56 | 52.83 | 93.67 | -0.31 |
| LSC(Lee et al., 2020) | ✓ | ResNet56 | 55.45 | 93.39 | -0.23 |
| ACP(Chang et al., 2022) | ✓ | ResNet56 | 54.42 | 93.18 | +0.21 |
| REPrune(Park et al., 2024) | × | ResNet56 | 60.38 | 93.39 | +0.01 |
| **SPTE** | ✓ | **ResNet18(SNN)** | **59.21** | **94.19** | **-0.88** |

10 dataset as an example, we observe that as the pruning ratio increases, the number of channels in each layer decreases proportionally; moreover, the reduction is considerably more pronounced in deep layers than in shallow ones, and at higher pruning ratios, some deep networks retain even fewer channels than shallow layers. This trend suggests that deeper layers contain a higher proportion of redundant or low-sensitivity channels, making them ideal candidates for more aggressive pruning without significantly compromising model performance. In contrast, shallow layers, being closer to the input, are essential for capturing fine-grained features such as colors, textures, edges, and corners. Although deeper layers benefit from larger receptive fields and richer semantic representations, they exhibit reduced sensitivity to detailed information, resulting in a lower contribution of certain neurons to the final output and further validating their suitability for pruning.

### 4.3 Performance with Different Timesteps

To investigate the effect of different time steps on SPTE, we conduct ablation experiments on the T. Table 2 presents the performance of SNN VGG16 on dataset CIFAR10 with time steps of 2, 4, and 8. It can be seen that when the pruning ratio reaches 1/2, the accuracy of time steps 2 and 4 decreases by about 1%. This may be due to instability caused by a short time step. However, the decrease in parameter count and FLOPs is stable, maintaining around 63% and 16%, respectively. It can be observed that SPTE is relatively stable for different time steps.

### 4.4 Comparison with Other Methods

**SNN Method.** Table 3 presents a comparative analysis of our method against existing SNN pruning approaches. In order to make the comparison more comprehensive, we add average synaptic operations and parameter count corresponding to different accuracy losses.

On the CIFAR-10 dataset with ResNet18, our method retains more SOPs compared to other method (Chen et al., 2021; Shen et al., 2023; Shi et al., 2024; Wei et al., 2025)[1], but our accuracy loss is relatively low. Although Unstruct-Pru (Shi et al., 2024)[2] has lower SOPs, their method is unstructured pruning, which is not hardware friendly. In contrast to gradient ratio-based pruning (Grad R) (Chen et al., 2021), SPTE may not achieve the most aggressive parameter compression, but it offers distinct advantages: (1) we apply structured pruning directly to convolutional layers and require only light fine-tuning to recover or even improve model performance, thereby avoiding costly retraining from scratch; and (2) redundant structural units are completely removed, resulting in a more compact and hardware-friendly architecture.

On the CIFAR-100 dataset with Resnet18, our accuracy loss and parameters are both less than Unstruct-Pru (Shi et al., 2024).For the Tiny ImageNet dataset, when both SOPs and parameters are reduced, the accuracy of our method improves significantly. We also evaluated our approach on neuromorphic datasets, which have received less attention in prior work. On the DVS-CIFAR10 dataset, although the parameters remained is relatively more, the SOPs is substantially reduced, and accuracy decreased slightly, even increases at lower pruning rotio.

---

[1]The data of ESL-SNNs method we use comes from Shi et al. (2024).

[2]For ease of description, we refer to the method of Shi et al. (2024) as 'Unstruct-Pru'.

Overall, these results confirm that our SPTE-based structured pruning approach effectively identifies and removes redundant structural units, enabling the construction of compact and regularized networks. With minimal fine-tuning, the pruned models can quickly recover or even surpass the original accuracy, while being easier to deploy on hardware platforms compared to unstructured pruning techniques (Shrestha et al., 2022).

**ANN Method.** Table 4 presents a comparative analysis of our method against existing ANN pruning approaches. 'PT?' indicates whether a method necessitates pre-training the original model as part of the train-pruning-finetuning pipeline (✓) or if it adheres to a concurrent training-pruning paradigm (×). It can be observed that our method has an accuracy loss of only 0.88% with FLOPs reduction rates of 59.21%. Although there is more accuracy loss compared to ANN, this is more due to the limitations brought by SNN itself.

## 5 CONCLUSION

Lightweight, high-performance SNNs not only benefit from reduced power consumption but also achieve faster inference, enabling real-time applications. Structural pruning produces regular, sparse models that are more compatible with hardware accelerators. Our approach begins with sensitivity analysis, evaluating the importance of each channel to determine optimal pruning strategies. This enables precise and efficient parameter reduction, compressing the network while preserving high performance. Such a method is particularly valuable for deploying high-performance, memory-efficient SNNs on neuromorphic chips.

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

# A APPENDIX

## A.1 USE OF LLMs

Large Language Models(LLMs) were used solely to assist with polishing the text.

## A.2 CODE OF ETHICS AND ETHICS STATEMENT

The research conducted in the paper conform, in every respect, with the ICLR Code of Ethics https://iclr.cc/public/CodeOfEthics.

Table 5: The energy between ANN and SNN.

| Dataset | Method | Ratio | MAC(M) | AC(M) | Energy(mJ) |
|---------|--------|-------|--------|-------|------------|
| CIFAR10 | VGG16(ANN) | 0 | 333.218 | - | 1.5328 |
| | VGG16(SNN) | 0 | - | 50.484 | 0.0454 |
| | | 3/4 | - | 43.359 | 0.0390 |
| CIFAR100 | RenNet18(ANN) | 0 | 557.935 | - | 2.5665 |
| | RenNet18(SNN) | 0 | - | 328.531 | 0.2957 |
| | | 3/4 | - | 163.940 | 0.1475 |

## A.3 THE BENEFIT OF SNN IN ENERGY

In order to demonstrate the advantages of SNN on neuromorphic chips, we referred to Horowitz (2014) and calculated the theoretical energy consumption of ANN and SNN under the same structure. We present in Table 5.

$$Power(ANN) = (3.7 + 0.9)pJ \times MAC \tag{8}$$

$$Power(SNN) = 0.9pJ \times AC \tag{9}$$

For VGG16 on CIFAR10, it can be seen that the energy consumption of SNNs with the same structure is only 2.96% of ANN, and when the pruning ratio is 3/4, it is only 2.54%.

