# OpenReview forum: "Towards Efficient SNNs: Sensitivity-Guided Pruning for Deep Spiking Architectures"
_ICLR.cc/2026/Conference — Submitted to ICLR 2026_

### Official Review · Reviewer_cwzv · 2025-10-26

**Soundness:** 1
**Presentation:** 2
**Contribution:** 1
**Rating:** 2
**Confidence:** 5

**Summary:**

This work proposes a Taylor-expansion-based structured pruning method for SNNs, and validates its performance on static and neuromorphic datasets.

**Strengths:**

1. In addition to the inference accuracy after pruning, this work also conducted statistical analysis on inference speed.

**Weaknesses:**

1. As shown in Tab.2, the performance comparison for this work is too simple:
- Firstly, some works that maintain superior performance on SNNs with extremely high sparsity have not been included.
- Secondly, this work did not include Synaptic Operations (SOPs), which is an important indicator for evaluating SNN power consumption.
- Thirdly, the results presented in this work are all under conditions of low sparsity (Connectivity > 50%), without verifying the performance of the method under conditions of extremely high sparsity. Some cases even experienced accuracy loss (Acc. Loss < 0%) under the conditions of Connectivity > 50%.
- In addition, this work only conducted performance validation on convolutional architectures and did not conduct performance validation on large-scale datasets (e.g. ImageNet-1k).
- The authors should consider using a network structure consistent with the comparative works for clear and detailed performance comparison. Meanwhile, the inference accuracies achieved in this work is clearly unsatisfactory in the current SNN community.

2. Pruning towards the synaptic layers is just one solution to reduce the power consumption of SNNs. Other techniques such as pruning for neuron layers and lightweight quantization for SNNs have also been proposed. Therefore, I tend to think the contribution of this work to the SNN community is relatively limited.

3. The layout of figures, tables and formulas in this paper still needs further polishing.

[1] Towards Energy Efficient Spiking Neural Networks: An Unstructured Pruning Framework. ICLR 2024.

[2] QP-SNN: Quantized and Pruned Spiking Neural Networks. ICLR 2025.

**Questions:**

See Weaknesses Section.

---

> ### Author Response · Authors · 2025-11-28
> **Response to  Reviewer cwzv**
>
> We are grateful for the time and effort the reviewer dedicated to reviewing our manuscript. The feedback has been instrumental in refining our work, and we have made the necessary revisions.
>
> **Q1: The performance comparison for this work is too simple.**
>
> R1:Thank you for your suggestions on the shortcomings of our work.
>
> 1.	We have added the two methods you mentioned[3][4] in Table 3. For ease of description, we refer to the method of [3] as 'Unstruct-Pru'. On CIFAR10, we outperform Unstruct-Pru in both accuracy loss and parameter compression. On CIFAR100, compared with QP-SNN, although our accuracy drops more, the parameter compression ratio is higher than theirs. On DVS-CIFAR10, our baseline model may not be very good, but the accuracy loss will be lower than Unstruct-Pru and QP-SNN. On Tiny-ImageNet, our performance is superior to QP-SNN in terms of both accuracy loss and parameter compression.
>
> 2.	We have added Synaptic Operations (SOPs) in Table 3 and replaced the original FLOPs with SOPs in Figure 2. As seen from the red line in Figure 2, with an increase in the network’s channel pruning ratio, the synaptic operations gradually decrease.
>
> 3.	To further compare the performance of the model, we have supplemented the data with Connectivity < 50, as shown in Figure 2. In our work, 'Connectivity' refer to the weight left, corresponding to the parameter. This is different from [3], which refers topercentageof synaptic connections left.
>
> 4.	We have added datase Food-101 in Figure 2(e). Food-101 contains 101 food categories with a total of 101000 images. Each category provides 250 manually reviewed test images and 750 training images. All images are scaled to a maximum edge length of 512 pixels. We resize to $224\times224$ encoded with 4 timesteps.
>
> | Dataset | Method | Network | T | Top-1 Acc. (%) | Acc.Loss (%) | Avg.SOPs (M) | Param. (M) |
> |---------|--------|---------|---|----------------|--------------|--------------|------------|
> | CIFAR10 | Grad R[1] | 6Conv+2FC | 8 | 92.84 | -0.30 | 371.05 | 10.43 |
> |  |  |  |  |  | -0.81 | 143.69 | 1.86 |
> |  |  |  |  |  | -3.52 | 54.89 | 0.26 |
> |  | ESL-SNNs[2] | ResNet19 | 2 | 92.71 | -0.63 | 298.24 | 2.53 |
> |  |  |  |  |  | -1.25 | 178.10 | 1.26 |
> |  |  |  |  |  | -1.81 | 108.89 | 0.63 |
> |  | Unstruct-Pru[3] | 6Conv+2FC | 8 | 92.84 | -0.21 | 38.32 | 12.57 |
> |  |  |  |  |  | -0.79 | 16.47 | 9.56 |
> |  |  |  |  |  | -2.19 | 8.50 | 7.10 |
> |  | QP-SNN[4] | ResNet20 | 2 | 95.12 | -0.56 | - | 3.92 |
> |  | **SPTE** | ResNet18 | 8 | 94.19 | -0.36 | 535.19 | 7.17 |
> |  |  |  |  |  | -0.88 | 441.38 | 5.31 |
> |  |  |  |  |  | -1.31 | 272.70 | 2.59 |
> | CIFAR100 | Unstruct-Pru[3] | Resnet18 | 4 | 74.16 | -1.82 | 27.16 | 13.18 |
> |  |  |  |  |  | -2.85 | 15.54 | 9.86 |
> |  |  |  |  |  | -3.71 | 9.60 | 7.67 |
> |  | QP-SNN[4] | ResNet20 | 2 | 75.29 | -0.51 | - | 4.10 |
> |  | **SPTE** | Resnet18 | 4 | 74.31 | -0.59 | 372.88 | 9.24 |
> |  |  |  |  |  | -1.88 | 284.88 | 6.40 |
> |  |  |  |  |  | -3.05 | 212.58 | 4.04 |
> | DVS-CIFAR10 | ESL-SNNs[2] | VGGSNN | 10 | 82.4 | -1.8 | 266.17 | 1.94 |
> |  |  |  |  |  | -3.2 | 152.64 | 0.97 |
> |  |  |  |  |  | -4.9 | 79.65 | 0.48 |
> |  | Unstruct-Pru[3] | VGGSNN | 10 | 82.4 | -0.5 | 47.81 | 3.52 |
> |  |  |  |  |  | -3.4 | 10.02 | 2.18 |
> |  |  |  |  |  | -4.1 | 6.75 | 1.81 |
> |  | **SPTE** | 4Conv+2FC+Vote | 4 | 71.60 | +0.3 | 66.75 | 17.33 |
> |  |  |  |  |  | +0.5 | 31.44 | 13.05 |
> |  |  |  |  |  | -0.1 | 22.24 | 10.66 |
> | Tiny-ImageNet | SCA-based[5] | VGG16 | 4 | 49.33 | +0.03 | - | 27.92 |
> |  |  |  |  |  | -0.19 | - | 19.76 |
> |  | QP-SNN[4] | VGG16 | 2 | 53.32 | -1.33 | - | 4.67 |
> |  | **SPTE** | VGG16 | 4 | 52.99 | +2.77 | 395.81 | 34.67 |
> |  |  |  |  |  | +1.1 | 198.18 | 25.14 |
> |  |  |  |  |  | -1.09 | 123.99 | 21.67 |
>
>
>
> **Q2:Pruning towards the synaptic layers is just one solution to reduce the power consumption of SNNs and the contribution is relatively limited.**
>
> R2:We emphasize that the core contribution of this article is to provide a principled and hardware friendly structured pruning paradigm, filling the gap in SNN pruning in the field of sensitivity analysis. The pruning at the neuron level mentioned in [1] is unstructured and not very hardware friendly, which is in opposition to our structured approach. The quantization method mentioned in [2] does indeed further compress the parameter count of the model, and we will consider incorporating quantization methods in future work.
>
> **Q3:The layout of figures, tables and formulas in this paper still needs further polishing.**
>
> R3:Thank you for your suggestion, we have re-polished the layout of figures and tables to ensure clarity and professionalism. In Figure 2, we have added data with higher sparsity and replaced the original FLOPs with SOPs. Meanwhile, we have added more tables for data display and comparison. In Table 3, we add SOPs and replace 'Connectivity' with 'Param.'.

---

> > ### Author Response · Authors · 2025-11-28
> > **Response to Reviewer cwzv**
> >
> > **Reference**
> >
> > [1]Pruning of deep spiking neural networks through gradient rewiring, 2021.
> > [2]Esl-snns: An evolutionary structure learning strategy for spiking neural networks, 2023.
> > [3]Towards Energy Efficient Spiking Neural Networks: An Unstructured Pruning Framework, ICLR 2024.
> > [4]QP-SNN: Quantized and Pruned Spiking Neural Networks,  ICLR 2025.
> > [5]Towards Efficient Deep Spiking Neural Networks Construction with Spiking Activity based Pruning, 2024.

---

### Official Review · Reviewer_Zx95 · 2025-10-30

**Soundness:** 2
**Presentation:** 2
**Contribution:** 2
**Rating:** 6
**Confidence:** 5

**Summary:**

The paper presents STE, an innovative structured pruning framework for SNNs, which effectively tackles the challenge of network compression. Extensive experiments highlight its promising capability to significantly reduce model size and FLOPs across various benchmark datasets.

**Strengths:**

1. Innovation Method: Hardware-Friendly Structured Pruning Framework. By using first-order Taylor expansion for sensitivity evaluation, STE performs channel- or kernel-level pruning that aligns with the spatiotemporal characteristics of SNNs.
2. Superior Compression–Accuracy Trade-off: STE achieves remarkable model compression while maintaining or even improving accuracy across multiple benchmarks.

**Weaknesses:**

1. The citation style is inconsistent and does not conform to the official ICLR formatting guidelines.
2. Related Work:
This section lacks depth and persuasiveness due to an insufficient number of relevant citations (which I think is incomprehensible). A qualified study needs to be supported and corroborated by previous papers. Expanding the discussion to include more recent and influential SNN pruning and optimization works would strengthen the context of this research.
3. The claim of “ensuring compatibility and performance retention across diverse architectures” (lines 069–070) appears overstated, given that experiments are restricted to VGG- and ResNet-style networks.
4. The results in Table 2 raise concerns: under identical architectures, the proposed STE method occasionally performs worse than competing pruning techniques.

**Questions:**

Please refer to “Weaknesses”

---

> ### Author Response · Authors · 2025-11-28
> **Response to  Reviewer Zx95**
>
> Thank you for your insightful and detailed comments. We have carefully considered each suggestion and made the appropriate revisions to enhance the manuscript.
>
> **Q1:The citation style is inconsistent and does not conform to the official ICLR formatting guidelines.**
>
> R1:We appreciate your careful corrections. We have strictly followed the ICLR format guidelines to unify all references in the paper.
>
> **Q2:Related Work: Insufficient number of relevant references.**
>
> R2:Thanks for your correction.We have added references to relevant work to increase the depth and persuasiveness of the article.
>
> **Q3:The claim of 'ensuring compatibility and performance retention across diverse architectures' is overstated.**
>
> R3:We agree that the wording can be more precise. We have modifidf this sentence to a more specific description: ' ensuring compatibility and performance retention on multiple convolutional network architectures such as VGG and ResNet'. And in the future, we will try other network architectures to verify the feasibility of our method, such as spikeformer.
>
> **Q4:The results in Table 2 show insufficient competitiveness.**
>
> R4: Thanks for your suggestion. We have supplemented the data for [1] and [2], and added two advanced method[3][4] to enhance our competitive standing.
>
> | Dataset | Method | Network | T | Top-1 Acc. (%) | Acc.Loss (%) | Avg.SOPs (M) | Param. (M) |
> |---------|--------|---------|---|----------------|--------------|--------------|------------|
> | CIFAR10 | Grad R[1] | 6Conv+2FC | 8 | 92.84 | -0.30 | 371.05 | 10.43 |
> |  |  |  |  |  | -0.81 | 143.69 | 1.86 |
> |  |  |  |  |  | -3.52 | 54.89 | 0.26 |
> |  | ESL-SNNs[2] | ResNet19 | 2 | 92.71 | -0.63 | 298.24 | 2.53 |
> |  |  |  |  |  | -1.25 | 178.10 | 1.26 |
> |  |  |  |  |  | -1.81 | 108.89 | 0.63 |
> |  | Unstruct-Pru[3] | 6Conv+2FC | 8 | 92.84 | -0.21 | 38.32 | 12.57 |
> |  |  |  |  |  | -0.79 | 16.47 | 9.56 |
> |  |  |  |  |  | -2.19 | 8.50 | 7.10 |
> |  | QP-SNN[4] | ResNet20 | 2 | 95.12 | -0.56 | - | 3.92 |
> |  | **SPTE** | ResNet18 | 8 | 94.19 | -0.36 | 535.19 | 7.17 |
> |  |  |  |  |  | -0.88 | 441.38 | 5.31 |
> |  |  |  |  |  | -1.31 | 272.70 | 2.59 |
> | CIFAR100 | Unstruct-Pru[3] | Resnet18 | 4 | 74.16 | -1.82 | 27.16 | 13.18 |
> |  |  |  |  |  | -2.85 | 15.54 | 9.86 |
> |  |  |  |  |  | -3.71 | 9.60 | 7.67 |
> |  | QP-SNN[4] | ResNet20 | 2 | 75.29 | -0.51 | - | 4.10 |
> |  | **SPTE** | Resnet18 | 4 | 74.31 | -0.59 | 372.88 | 9.24 |
> |  |  |  |  |  | -1.88 | 284.88 | 6.40 |
> |  |  |  |  |  | -3.05 | 212.58 | 4.04 |
> | DVS-CIFAR10 | ESL-SNNs[2] | VGGSNN | 10 | 82.4 | -1.8 | 266.17 | 1.94 |
> |  |  |  |  |  | -3.2 | 152.64 | 0.97 |
> |  |  |  |  |  | -4.9 | 79.65 | 0.48 |
> |  | Unstruct-Pru[3] | VGGSNN | 10 | 82.4 | -0.5 | 47.81 | 3.52 |
> |  |  |  |  |  | -3.4 | 10.02 | 2.18 |
> |  |  |  |  |  | -4.1 | 6.75 | 1.81 |
> |  | **SPTE** | 4Conv+2FC+Vote | 4 | 71.60 | +0.3 | 66.75 | 17.33 |
> |  |  |  |  |  | +0.5 | 31.44 | 13.05 |
> |  |  |  |  |  | -0.1 | 22.24 | 10.66 |
> | Tiny-ImageNet | SCA-based[5] | VGG16 | 4 | 49.33 | +0.03 | - | 27.92 |
> |  |  |  |  |  | -0.19 | - | 19.76 |
> |  | QP-SNN[4] | VGG16 | 2 | 53.32 | -1.33 | - | 4.67 |
> |  | **SPTE** | VGG16 | 4 | 52.99 | +2.77 | 395.81 | 34.67 |
> |  |  |  |  |  | +1.1 | 198.18 | 25.14 |
> |  |  |  |  |  | -1.09 | 123.99 | 21.67 |
>
> **Reference**
>
> [1]Pruning of deep spiking neural networks through gradient rewiring, 2021.
> [2]Esl-snns: An evolutionary structure learning strategy for spiking neural networks, 2023.
> [3]Towards Energy Efficient Spiking Neural Networks: An Unstructured Pruning Framework, ICLR 2024.
> [4]QP-SNN: Quantized and Pruned Spiking Neural Networks,  ICLR 2025.
> [5]Towards Efficient Deep Spiking Neural Networks Construction with Spiking Activity based Pruning, 2024.

---

### Official Review · Reviewer_wA6y · 2025-10-31

**Soundness:** 3
**Presentation:** 3
**Contribution:** 2
**Rating:** 4
**Confidence:** 4

**Summary:**

The paper proposes ​​STE (Sensitivity-guided pruning by Taylor Expansion)​​, a structured pruning framework for Spiking Neural Networks (SNNs). STE uses Taylor expansion to estimate the sensitivity of each convolutional kernel to the loss function, guiding the pruning process to preserve important structures for efficient execution on general-purpose hardware.

**Strengths:**

The focus on ​​structured pruning​​ is a benefit, as it produces hardware-friendly models that can run efficiently on standard GPUs and CPUs without the need for specialized sparse acceleration hardware.

The method appears to be straightforward and simple, building on the established concept of Taylor expansion for importance estimation. This makes it potentially easy to understand and implement.

By estimating sensitivity during the training process, the method can dynamically adapt the pruning strategy, potentially leading to better preservation of accuracy compared to one-shot pruning methods.

**Weaknesses:**

A major concern is that the core framework of the proposed method may not be sufficiently novel, potentially being an incremental application of existing ideas to ANNs. The use of channel/kernel sensitivity evaluation via Taylor expansion is a known technique in the ANN literature, and the paper may not demonstrate enough adaptation or innovation to make it compelling for the SNN domain.

Table 2 compares the method against too few existing state-of-the-art benchmarks, making it difficult to assess its true competitive standing.

The citation format is problematic (e.g., "Recent studies Lietal. (2024a;b)"). Please use \citep appropriately.
Sections 2.1 and 2.2 lack citations entirely, and 2.3 has too few, weakening the literature review.

The use of "So" at the beginning of a sentence (Line 191) is inappropriate for academic writing.

The acronym "STE" is already widely used in the SNN field for "Straight-Through Estimator," which is the standard method for training SNNs. This creates immediate and significant confusion.

**Questions:**

What is the specific novel contribution of this work that differentiates it from simply applying existing ANN structured pruning techniques to SNNs?

The method is demonstrated on convolutional architectures. Can the proposed framework be effectively applied to more modern, Transformer-like SNN architectures?

The paper reports inference speed on a GPU. However, a key motivation for using SNNs is their efficiency on neuromorphic hardware. What are the expected benefits or performance of the pruned models on neuromorphic processors, and why was this not evaluated?

Given the critical name conflict with "Straight-Through Estimator (STE)," would the authors consider changing the name of their method to avoid confusion and improve the identity of their work?

---

> ### Author Response · Authors · 2025-11-28
> **Response to  Reviewer wA6y**
>
> We would like to thank the reviewer for the thoughtful and constructive comments. The suggestions provided have significantly improved the clarity and quality of the manuscript.
>
> **W1 and Q1:What is the specific novel contribution of this work?**
>
> The specific novel contribution of this work is the introduction of a Taylor-expansion-based sensitivity analysis framework for structured pruning in convolutional SNNs, by combining with the intrinsic spatiotemporal property of SNNs. Unlike traditional pruning methods in ANNs [1][2], our approach is not a direct adaptation of ANN-based techniques. While pruning in ANNs typically relies on magnitude and similarity-based methods that focus on weights, filters, and channel attributes, these approaches do not take into account the unique dynamics of SNNs.
>
> Our method, on the other hand, specifically targets the kernel-level importance of spiking units, which is essential for making informed pruning decisions in SNNs. It uses sensitivity analysis to model the impact of perturbation weights on the loss function in a way that aligns with the spiking nature of neurons in SNNs. Although sensitivity analysis has been applied in ANNs, it has not been successfully adapted to SNNs, and this work fills that gap by developing a framework tailored to the specific characteristics of spiking neural networks.
>
>
> **W2:Table 2 Lack of Competition.**
>
> R2: Thanks for your suggestion. We have supplemented the data for [3] and [4], and added two advanced method[1][5] to enhance our competitive standing in the Table 3 in the revision.
>
>
> **W3:Citation format issues and lack of citations.**
>
> R3:Thanks for your correction. We have made corrections to 'Recent studies Lietal. (2024a; b)' and added relevant references in Sections 2.
>
> **W4:The use of "So" at the beginning of a sentence.**
>
> R4: Thanks for your correction. We have replaced 'So' with 'Therefore'.
>
> **W5 and Q4:Conflict with the name of 'Straight Through Estimation (STE)'.**
>
> R5:We thank the reviewer for this important feedback. To avoid any confusion with the well-known "Straight-Through Estimator (STE)" in SNN training, we have renamed our method from "STE" to "SPTE" (Sensitivity-guided Pruning by Taylor Expansion) throughout the entire manuscript.
>
> **Q2:Can the proposed framework be effectively applied to more modern, Transformer-like SNN architectures?**
>
> R6:Thanks for your suggestion. Spiking transformers are indeed the mainstream network architecture currently, we will make this an important research direction in the future. We would like to emphasize that the core principle of our proposed method is fundamentally general and not limited to a specific network architecture. Because it is based on the gradient of activation using a loss function, which is independent of the specific type of neuron.
>
> **Q3:What are the expected benefits or performance of the pruned models on neuromorphic processor?**
>
> R7:Thank you for emphasizing this important aspect. Due to practical limitations in the laboratory, we regret to inform you that we do not yet have the conditions to use neuromorphic chips. In order to demonstrate the advantages of SNN on neuromorphic chips, we referred to [6] and calculated the theoretical energy consumption of ANN and SNN under the same structure. For VGG16 on CIFAR10, it can be seen that the energy consumption of SNNs with the same structure is only 2.96\% of ANN, and when the pruning ratio is 3/4, it is only 2.54\%.(The Ratio in the table represents prunned ratio.)
> \begin{equation}
>         Power(ANN)=(3.7+0.9)pJ\times MAC
>     \end{equation}
> \begin{equation}
>         Power(SNN)=0.9pJ\times AC
>     \end{equation}
>
> | Dataset   | Method         | Ratio | MAC(M) | AC(M)   | Energy(mJ) |
> |-----------|----------------|-------|--------|---------|------------|
> | CIFAR10   | VGG16(ANN)     | 0     | 333.218 | -       | 1.5328     |
> |    | VGG16(SNN)     | 0     | -      | 50.484  | 0.0454     |
> |    | VGG16(SNN)     | 3/4   | -      | 43.359  | 0.0390     |
> | CIFAR100  | RenNet18(ANN)  | 0     | 557.935 | -       | 2.5665     |
> |   | RenNet18(SNN)  | 0     | -      | 328.531 | 0.2957     |
> |   | RenNet18(SNN)  | 3/4   | -      | 163.940 | 0.1475     |
>
> **Reference**
>
> [1]Towards Energy Efficient Spiking Neural Networks: An Unstructured Pruning Framework, ICLR 2024.
> [2]Towards Efficient Deep Spiking Neural Networks Construction with Spiking Activity based Pruning, 2024.
> [3]Pruning of deep spiking neural networks through gradient rewiring, 2021.
> [4]Esl-snns: An evolutionary structure learning strategy for spiking neural networks, 2023.
> [5]QP-SNN: Quantized and Pruned Spiking Neural Networks,  ICLR 2025.
> [6]Computing’s Energy Problem (and what we can do about it), 2014.

---

### Official Review · Reviewer_JBuL · 2025-11-01

**Soundness:** 3
**Presentation:** 3
**Contribution:** 2
**Rating:** 6
**Confidence:** 5

**Summary:**

This paper proposes a novel structured pruning framework for SNNs named STE (Sensitivity-guided pruning by Taylor Expansion). The method leverages first-order Taylor expansion to estimate the sensitivity of convolutional kernels to the loss function, enabling the iterative removal of less critical components in a structured manner. The pruned models show significant reductions in FLOPs and improvements in inference speed.

**Strengths:**

1. The paper successfully adapts a principled, Taylor-expansion-based sensitivity analysis from ANNs to the SNN domain, addressing a significant gap in structured pruning for spiking architectures. The approach is well-motivated by the limitations of existing unstructured pruning methods.

2. The paper provides valuable insights beyond mere accuracy and sparsity numbers. The analysis of sensitivity distribution across layers (Figure 3) offers an intuitive explanation for why deeper layers can be pruned more aggressively, strengthening the methodological foundation.

**Weaknesses:**

1. While the paper compares favorably against other SNN pruning methods, it would be strengthened by a brief discussion or comparison with state-of-the-art structured pruning techniques applied to ANNs on the same tasks. This would help contextualize STE's performance within the broader model compression field.

2. The description of the iterative pruning process could be more detailed. Specifically, the total number of pruning iterations, the schedule for fine-tuning (e.g., number of epochs per iteration), and the associated computational cost are not explicitly stated.

**Questions:**

1. The method calculates sensitivity over the entire temporal dimension (T). How does the performance of STE change with a different number of timesteps? Is the sensitivity metric consistently reliable across varying temporal sequence lengths?

2. Can this approach be extended to other mainstream neural network frameworks like spiking transformers?

3. Can this approach be extended to more large-scale datasets/tasks?

4. Can this approach be extended to more non-vision tasks?

---

> ### Author Response · Authors · 2025-11-28
> **Response to  Reviewer JBuL**
>
> We sincerely appreciate the reviewer’s thorough review and valuable feedback. We have carefully addressed the points raised and revised the manuscript accordingly.
>
> **W1:Comparison with ANN structured pruning method.**
>
> R1: Thanks for the suggestion. Table 1 presents a comparative analysis of our method against existing ANN pruning approaches. It can be observed that our method has an accuracy loss of only 0.88\% with FLOPs reduction rates of 59.21\%. Although there is more loss compared to ANN, this is more due to the limitations brought by SNN itself. (‘PT?’ indicates whether a method necessitates pre-training the original model as part of the train-pruning-finetuning pipeline ($\checkmark$) or if it adheres to a concurrent training-pruning paradigm ($\times$).)
>
> | Method | PT? | Network | FLOPs reduction(%) | Acc.(%) | Acc.Loss(%) |
> |--------|-----|---------|-------------------|---------|------------|
> | CUP[1] | × | ResNet56 | 52.83 | 93.67 | -0.31 |
> | LSC[2] | ✓ | ResNet56 | 55.45 | 93.39 | -0.23 |
> | ACP[3] | ✓ | ResNet56 | 54.42 | 93.18 | +0.21 |
> | REPrune[4] | × | ResNet56 | 60.38 | 93.39 | +0.01 |
> | SPTE | ✓ | ResNet18(SNN) | 59.21 | 94.19 | -0.88 |
>
> **W2:More detailed description of the iterative pruning process.**
>
> R2:Thank you for your suggestion. In our experiment, we set q\% to $\frac{1}{24}$ and p\% to $\frac{3}{4}$, which means a total of 18 iterations were performed (or stopped when the model performance was severely compromised), with 20 epochs of retraining per iteration. And we have replaced the original FLOPs with SOPs in Figure 2 to better show the computational cost.
>
> **Q1:The method calculates sensitivity over the entire temporal dimension (T).**
>
> R3:We are very grateful for your valuable suggestions. We have added experiments on different timesteps about 2 and 4.  It can be seen that the sensitivity metric remains reliable across different temporal sequence lengths.
>
> | T | Ratio | Acc.(%)   | Parameter(M) | FLOPs(G) |
> |---|-------|-------|--------------|----------|
> | 2 | 0     | 91.27 | 33.64        | 193.63   |
> |   | 1/2   | 90.10 | 21.37        | 30.72    |
> | 4 | 0     | 92.14 | 33.64        | 387.26   |
> |   | 1/2   | 91.14 | 21.37        | 60.88    |
> | 8 | 0     | 90.58 | 33.64        | 774.51   |
> |   | 1/2   | 91.53 | 21.25        | 131.08   |
>
> **Q2: Can this approach be extended to other mainstream neural network frameworks like spiking transformers?**
>
> R4:Thanks for your question. Yes, this approach can be extended to other neural network frameworks including spiking transformers. We would like to emphasize that the core principle of our proposed method is fundamentally general and not limited to a specific network architecture. Because it is based on the gradient of activation using a loss function, which is independent of the specific type of neuron. We will make this an important research direction in the future.
>
> **Q3:Can this approach be extended to more large-scale datasets/tasks?**
>
> R5:Thanks for your suggestion. We have added datase Food-101 in Figure 2(e) in the revision. Food-101 contains 101 food categories with a total of 101000 images. Each category provides 250 manually reviewed test images and 750 training images. All images are scaled to a maximum edge length of 512 pixels. We resize to $224\times224$ encoded with 4 timesteps. As shown in Figure 2(e), When the pruning ratio is 1/12, the accuracy increases by 1.63\%, then shows a slow downward trend. When the pruning ratio overtake 7/12, the accuracy is lower than that of the baseline model.
>
> **Q4:Can this approach be extended to more non-vision tasks?**
>
> R6:We believe this is also feasible. [5] and [6] are both structured pruning of large models. They demonstrate that even in the field of NLP, which requires extremely high semantic information, channel level pruning is completely feasible and can maintain model performance. In the field of audio, channel pruning has been successfully used to compress these audio processing models[7], with the difference being the timing signals they input. However, SNN is precisely capable of processing temporal signals.
>
> **Reference**
>
> [1]Cup: Cluster pruning for compressing deep neural networks, 2021.
> [2]Filter pruning and re-initialization via latent space clustering, 2020.
> [3]Automatic channel pruning via clustering and swarm intelligence optimization for CNN, 2022.
> [4]Reprune: Channel pruning via kernel representative selection, 2024.
> [5] Well-Read Students Learn Better: On the Importance of Pre-training Compact Models, 2019.
> [6] Structured Pruning of Large Language Models,2020.
> [7] Compressing Deep Neural Networks for Efficient Speech Enhancement, 2021.

---

### Author Response · Authors · 2025-12-04
**Response Summary about Review comments, ratings, and our clarification explanation**

Dear Area Chair and all Reviewers,

Thanks to the area chair and all the reviewers for taking the time to review our paper and providing valuable constructive feedback. We have made various improvements in response to these suggestions, including adding new experiments, clarifying the content, and expanding related discussions. Our modifications and responses are summarized as follows. For detailed responses to each review comment, please refer to the point by point response section.

Reviewer JBuL, Rating: 6. Reviewer gave a very positive evaluation, focusing more on the details of the experiment and the scalability of the method. In the revised version, we add more details of the iterative pruning process in Section 3.5. Meanwhile, we add performance with different timestep in Table 2 and comparison with ANN method in Table 4. In terms of dataset expansion, we add the large-scale dataset Food-101 in Figure 2(e). For the scalability of the new framework and non-vision tasks, we have a detailed discussion in our response.

Reviewer wA6y, Rating: 4. Reviewer corrects many errors in the details of our paper, and we have made successive revisions in the revised version. For instance, our original name conflict with "Straight-Through Estimator (STE), we rename our method from "STE" to "SPTE" (Sensitivity-guided Pruning by Taylor Expansion). The reviewer also raises doubts about our innovation, he considers that this is simply applying existing ANN structured pruning techniques to SNN. In fact, in previous work, pruning was based on the methods of magnitude and similarity, which implicitly addressed this goal by using weights, filters, and channel attributes that affect accuracy. In contrast, the sensitivity analysis method we introduced aims to model the impact of perturbation weights on the loss function. Although sensitivity analysis methods already exist in ANN, they have not been successfully applied to SNN. For the lack of competitiveness in Table 3 and SNN advantages reflected, we have provided more detailed data in Table 3 and added energy consumption calculations in the appendix. Meanwhile, we also discussed the scalability of the new framework.

Reviewer Zx95, Rating: 6. Reviewer also makes corrections to some details in our paper, such as the standardization of citations and the specificityof description. Meanwhile, the reviewer points out the shortcomings of our relevant work citations. As with Reviewer 2, this reviewer considers that the competitiveness of Table 3 is insufficient. All of these have been corrected or supplemented in the revised paper.

Reviewer cwzv, Rating: 2. Reviewer's main concern is that the performance comparison in Table 3 is too simple. For this reason, we have made significant modifications to Table 3. Firstly, we include the two methods mentioned by the reviewer in the comparison of Table 3. Secondly, in order to make the comparison more comprehensive, we add columns ‘T’ and ‘Avg.SOPs (M)’, and change ‘Connectivity(%)’ to ‘Param. (M)’ to eliminate ambiguity. Finally, we conduct experiments with higher pruning ratio and include the data in the comparison. In terms of large-scale datasets, we provide detailed data on Food-101 in Figure 2(e). Due to the lack of comparative data, we are unable to present them in Table 3. As for the reviewer's view that pruning the synaptic layer is only one solution to reduce SNN power consumption and our method's contribution is relatively limited, we believe this is a one-sided viewpoint. We emphasize that the core contribution of this article is to provide a principled and hardware friendly structured pruning paradigm, filling the gap in SNN pruning in the field of sensitivity analysis. For the figures and tables in this paper, we have modified and enriched some of them.

We earnestly request the Area Chair to consider this work in a broader academic context and contribution dimension when evaluating it. In response to wA6y(Rating 4), we have revised the paper in detail according to his suggestions, analyzed the innovation of our method, and provided the energy consumption calculation of SNN. In response to cwzv (Rating 2), we strictly follow the questions he raised in Q1, which include adding the latest two methods in Table 3, calculating SOPs, conducting experiments with higher pruning rates, and supplementing experiments on the large-scale dataset Food-101. Also, we discuss in detail the contributions of our method and modify the corresponding figures and tables.

It fills the gap of SNN pruning in sensitivity analysis and can still maintain good performance on large-scale datasets such as Food-101. We believe it can bring new ideas for SNN pruning, instead of being limited to magnitude- and similarity-based methods.

Best regards,

Authors

---

### Meta-Review · Area_Chair_VuWs · 2026-01-06

**Summary:**

This paper introduces SPTE (Sensitivity-guided Pruning by Taylor Expansion), a structured pruning framework for Spiking Neural Networks. It leverages Taylor expansion to estimate each convolutional kernel's sensitivity to the loss function during training and removes less critical components.

The reviewers' common concerns lie in the insufficient comparison with state-of-the-art methods and whether the proposed method can be extended to other architectures, e.g., spiking transformers and other large-scale datasets/tasks, e.g., ImageNet-1k or non-vision tasks. Additionally, the paper has some formatting and writing issues.

**Reviewer Concerns:**

After rebuttal, some of the concerns have been addressed. The authors supplemented the paper with relevant work citations and compared the proposed method with more state-of-the-art methods.

However, some concerns remain outstanding. The comparison in Table 3 employs inconsistent architectures across different methods for the same task, making it impossible to conduct a fair comparison between them. As a result, SPTE lags behind some state-of-the-art methods, for example, ESL-SNNs on the DVS-CIFAR10 dataset. Furthermore, the scalability of the proposed method to other architectures, e.g., spiking transformers and other large-scale datasets/tasks, e.g., ImageNet-1k, has not been fully validated.

Reviewer wA6y concerns that the proposed method may not be sufficiently novel, potentially being an incremental application of existing ideas to ANNs. I believe this concern is justified. The novelty of the proposed SPTE compared to existing sensitivity evaluation via Taylor expansion in ANNs needs further clarification.

**Reviewer Scores:**

I believe the rebuttal is insufficient to raise the scores. The primary issues remain inadequately addressed. Therefore, I believe the scores remain unchanged.

---

### Decision · Program_Chairs · 2026-01-26

Reject